# The Transcription Factor SCX is a Potential Serum Biomarker of Fibrotic Diseases

**DOI:** 10.3390/ijms21145012

**Published:** 2020-07-16

**Authors:** Miguel Ramírez-Aragón, Fernando Hernández-Sánchez, Tatiana S. Rodríguez-Reyna, Ivette Buendía-Roldán, Gael Güitrón-Castillo, Carlos A. Núñez-Alvarez, Diego F. Hernández-Ramírez, Sergio A. Benavides-Suárez, Alexia Esquinca-González, Ana Lilia Torres-Machorro, Criselda Mendoza-Milla

**Affiliations:** 1Departamento de Investigación en Fibrosis Pulmonar, Instituto Nacional de Enfermedades Respiratorias Ismael Cosío Villegas, Calzada de Tlalpan 4502, Colonia Belisario Domínguez Sección XVI, Alcaldía Tlalpan, Mexico City 14080, Mexico; mira0989@hotmail.com (M.R.-A.); ivettebu@yahoo.com.mx (I.B.-R.); gael-17@hotmail.com (G.G.-C.); 2Departamento de Neuropatología Molecular, División de Neurociencias, Instituto de Fisiología Celular, Universidad Nacional Autónoma de Mexico, Mexico City 04510, Mexico; 3Departamento de Investigación en Virología y Micología, Instituto Nacional de Enfermedades Respiratorias Ismael Cosío Villegas, Calzada de Tlalpan 4502, Colonia Belisario Domínguez Sección XVI, Alcaldía Tlalpan, Mexico City 14080, Mexico; fhernndezsnchez@gmail.com; 4Departamento de Inmunología y Reumatología, Instituto Nacional de Ciencias Médicas y Nutrición Salvador Zubirán, Av. Vasco de Quiroga 15, Colonia Belisario Domínguez Sección XVI. Alcaldía Tlalpan, Mexico City 14080, Mexico; sofarodriguez@yahoo.com.mx (T.S.R.-R.); nuac80df@gmail.com (C.A.N.-A.); diherram@gmail.com (D.F.H.-R.); benavidessergio.31@gmail.com (S.A.B.-S.); alexia_031@hotmail.com (A.E.-G.); 5Consejo Nacional de Ciencia y Tecnología and Instituto Nacional de Enfermedades Respiratorias Ismael Cosío Villegas, Calzada de Tlalpan 4502, Colonia Belisario Domínguez Sección XVI, Alcaldía Tlalpan, Mexico City 14080, Mexico

**Keywords:** bHLH transcription factor, idiopathic lung fibrosis, systemic sclerosis, TCF3

## Abstract

Fibrosing diseases are causes of morbidity and mortality around the world, and they are characterized by excessive extracellular matrix (ECM) accumulation. The bHLH transcription factor scleraxis (SCX) regulates the synthesis of ECM proteins in heart fibrosis. SCX expression was evaluated in lung fibroblasts and tissue derived from fibrotic disease patients and healthy controls. We also measured SCX in sera from 57 healthy controls, and 56 Idiopathic Pulmonary Fibrosis (IPF), 40 Hypersensitivity Pneumonitis (HP), and 100 Systemic Sclerosis (SSc) patients. We report high SCX expression in fibroblasts and tissue from IPF patients versus controls. High SCX-serum levels were observed in IPF (0.663 ± 0.559 ng/mL, *p* < 0.01) and SSc (0.611 ± 0.296 ng/mL, *p* < 0.001), versus controls (0.351 ± 0.207 ng/mL) and HP (0.323 ± 0.323 ng/mL). Serum levels of the SCX heterodimerization partner, TCF3, did not associate with fibrotic illness. IPF patients with severely affected respiratory capacities and late-stage SSc patients presenting anti-topoisomerase I antibodies and interstitial lung disease showed the highest SCX-serum levels. SCX gain-of-function induced the expression of alpha-smooth muscle actin (α-SMA/ACTA2) in fibroblasts when co-overexpressed with TCF3. As late and severe stages of the fibrotic processes correlated with high circulating SCX, we postulate it as a candidate biomarker of fibrosis and a potential therapeutic target.

## 1. Introduction

The excessive accumulation of extracellular matrix (ECM) components, such as collagen and fibronectin, in and around inflamed or damaged tissue defines fibrosis. This normal process is necessary for tissue repair; however, it can lead to permanent scarring and organ malfunction under pathological conditions. The cytokine TGFβ is a central mediator of fibrogenesis commonly upregulated and activated in fibrotic diseases [1]. TGFβ modulates fibroblast function and induces myofibroblast transdifferentiation in the process of fibrotic tissue remodeling [1,2].

Idiopathic Pulmonary Fibrosis (IPF) is a chronic and progressive Interstitial Lung Disease (ILD) of unknown etiology. It is characterized by persistent and progressive scarring of the lung parenchyma that ultimately leads to complete loss of pulmonary function. The prognosis is usually poor, with a median survival of 2 to 5 years without antifibrosis treatment [3]. Accurate diagnosis made by multidisciplinary teams of pulmonologists, radiologists, and pathologists sometimes requires invasive surgical lung biopsy. Despite bountiful research in the field, the natural history of the disease is variable, and progression remains unpredictable [3]. Hypersensitivity pneumonitis (HP) is also an ILD caused by inhalation of organic antigens that lead to diffuse immunopathologic reactions in the small airways and pulmonary parenchyma [4]. Approximately 10% of patients with continuous exposure develop chronic HP that may lead to lung fibrosis in a high number of patients.

Fibrosis is also a major pathological feature of several chronic autoimmune diseases such as Systemic Sclerosis (SSc) [5]. This disease is characterized by skin fibrosis, microvascular damage, and dysregulation of innate and adaptive immunity [6]. Clinical manifestations are heterogeneous, with a high percentage of patients developing generalized fibrosis in multiple organs, including the lung, with interstitial lung disease as the leading cause of death in these patients. Diagnosis remains challenging in patients with mild disease, or at early stages in the natural history of the disease.

Available biomarkers to aid the diagnosis and prognosis of fibrotic diseases are scant [7,8,9,10]. Examples include candidates informative of IPF susceptibility (MUC5B polymorphism and telomere shortening), diagnosis (MMP7 and CXCL13), differential diagnosis (MMP28) [11], and prognosis (KL6 and MMP7). Likewise, CXCL4, IL-6, IL-10, MMP9, and MMP12 are candidate biomarkers for SSc [12]. 

The basic helix-loop-helix (bHLH) transcription factor scleraxis (SCX) is involved in the embryonic development and function of ECM-rich tissues, including tendons and cardiac valves [13,14,15]. SCX actively participates in repairing and remodeling processes that require extracellular matrix synthesis [14,15]. However, defects in its regulatory function may lead to fibrosis in several organs, including the heart [16,17]. In a similar way to matrix proteins like fibronectin and the collagenase MMP2 [18,19], TGFβ can regulate the SCX expression [17,20]. SCX has been reported as a crucial regulator of the maintenance of the fibroblast phenotype and is necessary for the epithelial to mesenchymal transition (EMT). SCX downregulation attenuates myofibroblast function and gene expression [21]. Likewise, in SCX knockout mice, the cardiac matrix is deficient, with evidence of impaired EMT, and with ~50% loss of cardiac fibroblasts [21].

As SCX is an extracellular matrix regulator with a crucial role in fibroblast function, we hypothesized that its abnormal regulation could be involved in pathological fibrogenesis of diseases such as IPF and SSc. In this manuscript, we show that SCX expression was elevated in primary cell cultures and lung biopsies from patients with IPF in comparison to controls. Relative SCX concentrations in blood serum remained low in healthy control subjects and HP patients, whereas patients with IPF and SSc presented high circulating SCX. Furthermore, severe IPF and late SSc cases had the highest circulating SCX levels, supporting SCX as a candidate fibrosis biomarker of severity. 

## 2. Results

### 2.1. SCX Expression was High in Cells from Patients with Fibrotic Diseases

SCX expression was measured in the non-fibrotic cell lung lines CCD25Lu and CCD8Lu. SCX mRNA levels were higher in CCD8Lu fibroblasts in comparison with CCD25Lu (Figure 1a). Next, we tested the expression of SCX profibrotic target genes, finding that mRNA levels of genes encoding collagen (COL1A1 and COL1A2) and alpha-smooth muscle actin (α-SMA/ACTA2) were higher in CCD8Lu (Figure 1), the cell line that expressed more SCX.

Due to its role in regulating the expression of profibrotic genes, we expected SCX expression to be higher in primary lung fibroblasts from patients with pulmonary fibrosis. Thus, we measured SCX in three different cultures derived from patients with IPF (characteristics of donors are in Appendix A) versus primary fibroblasts from a healthy subject (NOVA). 

SCX expression levels were variable; however, all three IPF lines showed higher levels of SCX against the control line (Figure 2a). Similar to the expression pattern found in CCD25Lu and CCD8Lu, high SCX expression in IPF fibroblasts led to higher expression of SCX target genes COL1A1, COL1A2, and α-SMA/ACTA2 (Figure 2b–d). This increase did not correlate directly with SCX levels, as in Figure 1. For example, HIPF231 fibroblasts with the highest SCX expression showed the lowest increase in expression of target genes, relative to the other IPF cultures. In contrast, HIPF375 and HIPF397 showed a higher increase in target genes expression despite expressing lower SCX levels in comparison to HIPF231. Because functional SCX requires heterodimerization with E2A proteins [22,23], SCX expression was compared to TCF3 expression, which encodes the transcription factor E47 (Figure 2e). TCF3 levels were also variable among cultures: HIPF231 had about 25 times more SCX compared to TCF3, whereas HIPF375 and HIPF397 showed closer ratios of SCX/TCF3 relative mRNA levels (two- and sixfold, respectively). This fact supports previous findings that SCX function as a transcription factor is more efficient when adequate levels of its heterodimerization partner E47 allow structuring of the functional dimer [22,23]. Considering this, lines HIPF375 and HIPF397 showed high levels of expression of profibrotic genes COL1A1, COL1A2, and α-SMA/ACTA2. 

### 2.2. SCX Expression and Localization was Different in Patients with IPF Compared to Controls

As the gene expression in fibroblasts in culture could be different from that observed in tissue, we measured SCX gene expression directly in biopsy-derived RNA from IPF patients. Two commercial RNAs from healthy lung tissue were used as controls. We found high SCX expression in tissue from IPF patients relative to controls (Appendix A).

We, therefore, localized SCX by immunohistochemistry in lung tissue slides from both healthy controls and IPF patients. Our results revealed a positive label overall in the bronchiolar epithelium (Figure 3). Whereas, in IPF lungs, SCX was also present in fibroblast foci and inflammatory cells (Figure 3, samples 425-05, 428-16, and 442-16). 

The histological analyses confirmed differences in cell composition between normal and fibrotic lung. Healthy tissue was structured mainly by alveolar and muscle cells, whereas, in IPF tissue, alveolar spaces were replaced primarily by interstitial, mesenchymal origin cells, as previously reported [24]. This accumulation of SCX expressing cells could be responsible for SCX accumulation in IPF tissue. Thus, we hypothesized that SCX expression levels could correlate with the degree of fibrosis in patients with IPF. 

### 2.3. Circulating SCX was Increased in IPF and SSc Patients Compared to Healthy Subjects

Given that SCX expression was high in cells and tissue from patients with IPF, we tested if SCX serum levels were also augmented when compared to healthy controls. Circulating SCX was also measured in patients with two diseases with different forms of fibrosis: SSc and HP. All SSc patients have skin or internal organs fibrosis that may include the lung in up to 70% of patients, whereas fewer patients with HP develop pulmonary fibrosis. ELISA assays were used to measure SCX concentration in all cases. 

For this part of the study, we included 56 IPF patients, 40 HP patients, 100 SSc patients, and 57 age-matched controls. Table 1 summarizes the demographic and clinical characteristics of the study population. 

SCX serum levels were found significantly increased in both SSc (0.611 ± 0.296 ng/mL, *p* < 0.01) and IPF patients (0.663 ± 0.558 ng/mL, *p* < 0.001) compared to healthy controls (0.351 ± 0.207 ng/mL) (Figure 4). SCX concentration in sera from HP patients (0.323 ± 0.323 ng/mL) did not differ from controls.

Taking into consideration the existence of potential risk factors [25,26,27] and attempting to find other relevant associations, IPF patients were further categorized based on clinical data: smokers, non-smokers, ex-smokers, hypertensive [28], BMI, age, and sex (Figure 5). A comparison of SCX levels among those groups showed a significant increase in circulating SCX levels in non-smokers and hypertensive patients (*p* < 0.05) in contrast with the control group (Figure 5a). Non-obese IPF patients had significantly higher circulating SCX levels (Figure 5b) in comparison with controls, as well as male subjects and patients aged >61 years old (Figure 5c). The control groups did not show statistical differences in SCX concentrations when classified by demographic data (Appendix A).

IPF patients were also categorized based on pulmonary function tests (Figure 5d,e). Categories included Total Lung Capacity (TLC), classified as a severe decrease for patients with a capacity lower than 60%, as a moderate decrease for TLC between 61% and 69% and as acceptable for TLC over 70%. The Forced Vital Capacity (FVC) was classified into two categories: FVC below and over 80%. Groups with severely decreased TLC (≤60%) and FVC <80% showed significantly higher levels of serum SCX relative to controls (Figure 5d), suggesting an association between high SCX and late or severe fibrotic stages.

Similarly, patients with single-breath carbon monoxide diffusing capacity (DLCO) values below 75% showed also higher levels of circulating SCX (Figure 5e). Additionally, the classification of SCX serum levels based on the cellular composition of bronchoalveolar lavages (BAL) did not show statistical differences in any category (Appendix A). 

Clinical data grouping of SCX levels in SSc patients showed significant differences (*p* < 0.01) between healthy controls and SSc patients (mean 0.35 vs. 0.61 ng/mL). There were also statistically significant differences between groups of ex-smokers, non-smokers, and BMI < 25 in comparison with the control group (Figure 6a). The serological profile of SSc patients relative to controls showed the highest statistical difference in SCX levels in groups having anti-topoisomerase I antibodies and no anti-centromere antibodies (ACA) (Figure 6b). Even though most patients with SSc are women, higher SCX levels in serum were found in male patients (Figure 6c). 

Depending on the time of evolution of SSc, the progression is classified as early or late (see methods). The cohort in this study included patients belonging to both groups. Late-stage SSc patients showed the statistically highest difference in SCX levels compared to the control group. Diffuse and limited SSc variants also showed differences in SCX concentrations relative to controls but not between them (Figure 6e). Moreover, while in dcSSc patients (those with more severe disease) SCX levels remain high over time, there is a negative correlation between SCX levels and time of evolution in late lcSSc patients (*p* = 0.05, correlation coefficient −0.37).

SSc patients were also classified by internal organ involvement, including associated pulmonary fibrosis, pulmonary arterial hypertension, cardiac, gastrointestinal, renal, muscular, and joint involvement. The groups that showed statistically higher circulating SCX when compared to the healthy control group were pulmonary fibrosis, gastrointestinal involvement groups (GI), and arthritis (Figure 6d). The SCX values did not differ between FVC below or over 80% (Figure 6e). 

When the HP population was compared to the control group, it did not show any difference in SCX concentrations; however, it was statistically different from the SCX values in SSc and IPF groups (Figure 4). No statistical differences were discovered when the SCX concentration of HP patients was further categorized by clinical data (Appendix A). 

These results revealed higher SCX serum concentrations in patients with fibrotic diseases (IPF and SSc) versus healthy subjects. In patients with IPF, high SCX concentrations correlated with severe lung damage, measured through plethysmography and spirometry (TLC ≤ 60 and FVC < 80). In patients with SSc, SCX levels were increased in later stages of the disease and in patients with anti-topoisomerase I antibodies, pulmonary fibrosis (ILD-SSc), and gastrointestinal involvement. These data suggest that SCX is associated with the progression and severity of the fibrotic disease. 

### 2.4. Circulating TCF3 Levels were Similar among Controls, and IPF and SSc Patients 

Because SCX, as a functional transcription factor, heterodimerizes with E47 (TCF3), we measured TCF3 concentration in serum from patients and controls using an anti-TCF3 ELISA. We included seven IPF patients, eight HP patients, 21 SSc patients, and seven age-matched controls from the cohorts where circulating SCX was quantified. TCF3 levels were very high in all patients and healthy subjects, with all groups averaging values above 5 ng/mL (Figure 7). HP patients averaged the lowest circulating TCF3 (5.2 ± 2.6 ng/mL), whereas SSc patients had the highest average TCF3 serum levels (8.3 ± 4.2 ng/mL). Differences among tested groups were not statistically significant.

### 2.5. SCX Gain-of-Function Experiments Validated its Role in Lung Fibrosing Gene Expression

Elevated α-SMA/ACTA2 expression is a hallmark of fibrotic phenotype conversion. Thus, we evaluated the effect of SCX overexpression in CCD8Lu cells transduced with an adenoviral vector. SCX protein levels were highly induced 48 h after transduction (Figure 8). However, the expression of the target protein alpha smooth-muscle-actin (α-SMA/ACTA2) remained unchanged when compared to transduction with the empty vector. We reasoned that this result could be due to very low to absent TCF3 expression in CCD8Lu cells. Thus, TCF3 and SCX were co-overexpressed through the transduction of CCD8Lu cells, using independent TCF3 and SCX adenoviral vectors. This co-overexpression resulted in an induction of α-SMA/ACTA2 expression in comparison to cells transduced individually with the empty vector, SCX, or TCF3 (Figure 8). 

## 3. Discussion 

SCX is a matrix regulator, particularly for collagen gene expression in tendons, skin, and heart [13,14,15,17,21,30,31,32,33]. SCX expression has been reported in the lung without further characterization [34,35]. Herein, we found differences in SCX expression between two lung fibroblast cell lines (CCD25Lu and CCD8Lu). The fibroblast line expressing more SCX also showed higher levels of target genes COL1A1, COL1A2, and α-SMA/ACTA2, suggesting an SCX role in promoting target gene expression in lung fibroblasts. Because the cell line donors were aged 7 and 48 years old, respectively, age differences could influence the variable expression of the genes of interest. SCX expression changes throughout life: it is global during embryogenesis and becomes restricted to physical strength tissues like tendons during development and in adult organisms [36,37].

Abnormal expression of SCX plays a fundamental role in pathological processes in the heart and kidney [17,21,31]. For example, during mitral heart valve prolapse, high SCX levels promote the abnormal expression of proteoglycans and other ECM components [38]. Similarly, in cardiac fibrosis, the SCX dysregulation promotes exacerbated ECM synthesis, the formation of focal adhesions and tension fibers, and diminished migration and proliferation of cardiac cells [19,21,31,39]. In diabetic nephropathies, SCX also activates the expression of α-SMA/ACTA2 and bone morphogenetic protein 4, to promote differentiation of mesangial cells into activated myofibroblasts [40]. The above SCX roles in fibrotic diseases led us to analyze its role in pathological ECM deposition during lung fibrosis. When analyzing SCX expression in three primary fibroblast cell lines derived from patients with IPF, high levels were found in comparison with a normal fibroblast cell line. The expression of target genes was also high when compared to controls, but it did not correlate directly with SCX levels. When the expression of the gene encoding SCX heterodimerization partners (TCF3) was analyzed, we found variability in its mRNA levels. The ratio of SCX/TCF3 levels was very high for HIPF231, whereas HIPF375 and HIPF397 ratios were similar between them, and about five times lower in comparison to the HIPF231 rate. A previous report found that very high levels of SCX in an environment with low TCF3 levels did not result in high induction of SCX target genes [23].

Similarly, in our SCX gain-of-function experiments, SCX functionality and target gene induction were only observed when TCF3 was co-overexpressed in CCD8Lu cells, where basal TCF3 expression is deficient. HIPF231 expresses low TCF3 relative to SCX levels, correlating with the lowest activating capacity of target gene expression of all three IPF cultures tested. In the HIPF375 and HIPF395 fibrotic cell cultures, TCF3 levels were more balanced relative to SCX, and COL1A1, COL1A2, and α-SMA/ACTA2 expression were also more elevated. 

Considering that cultured primary fibroblasts from IPF could have altered expression due to in vitro culture conditions, we also tested SCX expression in RNA extracted directly from lung biopsies derived from patients with IPF. Again, SCX expression was higher in comparison with commercial tissue controls, which was in complete agreement with the histological data. Nevertheless, it remains unknown if the increased levels of SCX mRNA in the disease indicate higher expression of individual cells, higher fibroblast accumulation, or both. 

SCX expression in lung biopsies demonstrated that control and fibrotic lung tissue localized SCX expression to different cells: whereas it localized predominantly to the bronchiolar epithelium in the healthy lung, its presence was extended to fibroblast foci in biopsies from IPF patients. We propose that the accumulation of cells of mesenchymal origin in alveolar spaces could be responsible for high SCX expression in tissue from patients with pulmonary fibrosis. However, we cannot establish yet if SCX expression in these circumstances is a cause or a consequence of fibrosis.

Thus, high SCX expression in tissue and fibroblasts derived from IPF lungs in comparison to controls is characteristic of the disease. An example of another candidate bHLH tissue biomarker is ASCL2, a transcription factor involved in tumoral progression [41] that exhibits high abnormal expression in progressive cancer. Because of this, it was proposed as a prognosis biomarker in breast cancer, osteosarcoma, and lung squamous cell carcinoma [41,42,43]. In a similar way to ASCL2, the original SCX function was described as relevant in embryonic development; however, its high abnormal expression in tissue characterizes pulmonary fibrosis. SCX circulating levels were elevated in serum from IPF and SSc patients, whereas circulating protein levels in controls and patients with HP remained low. For both IPF and SSc, severe cases correlated with higher SCX levels in serum. IPF patients with critical pulmonary capacity loss: TLC ≤ 60% and FVC < 80% had higher SCX in serum. Lung function decline is related to the severity of IPF. Because we can measure circulating SCX in a noninvasive way, we propose SCX as a candidate biomarker of the severity of IPF. 

Furthermore, SSc patients had higher SCX levels than controls, particularly those in late stages of the disease, and with anti-topoisomerase I autoantibodies. As a systemic disease, in SSc, many factors have local and systemic effects on specific tissues that have been associated with severity and progression [44]. Circulating anti-topoisomerase I antibodies (Scl70) have been linked to diffuse disease and the presence and severity of interstitial lung diseases [45]. Scl70 is a poor prognosis marker due to the possibility of associated heart and kidney involvement as part of the fibrotic processes [46,47]. Patients with high SCX levels and anti-topoisomerase I antibodies usually do not have anti-centromere autoantibodies (ACA) in serum; these antibodies are mutually exclusive in the disease most of the time [48]. Thus, when ACA are present, the condition is usually milder, and there is a lower frequency of associated pulmonary fibrosis. Serological profile results are in complete agreement with organ involvement results, where patients with pulmonary fibrosis also had high SCX serum levels in comparison to controls. Therefore, in a similar way to IPF cases, high SCX in serum from SSc patients was also associated with severe cases with the worst prognosis. 

SSc patients with gastrointestinal involvement also had high SCX levels, probably reflecting the systemic involvement of the disease and the frequent visceral involvement; however, SCX levels did not correlate with the severity of gastrointestinal involvement. Even though SSc is more prevalent in women, men with the disease showed remarkably higher circulating SCX levels. This result could be related to the poorer prognosis and higher severity of the disease in male SSc patients when compared to female SSc patients [49]. Contrary to SSc, patients with IPF are predominantly males; however, in this condition, SCX levels were similar in women and men. 

SCX protein expression was analyzed in the murine pulmonary fibrosis model (induced by a single intratracheal instillation of bleomycin) [50]. We observed a nonsignificant trend towards higher SCX protein levels in the lungs from mice that developed pulmonary fibrosis (Appendix A). After finding high circulating SCX in patients with SSc, the murine model of this disease was also established [51]. Even though collagen levels in the skin increased considerably after 28 days of bleomycin treatment, SCX protein levels remained unchanged. Lungs from mice treated with subcutaneous bleomycin did not develop pulmonary fibrosis, nor increased SCX expression (Appendix A). That led us to suspect that circulating SCX in SSc could be derived from organ fibrosis, probably from lung, and not from skin fibrosis. IPF associated epithelial loss contrasts with skin fibrosis that is characterized by the accumulation of fibroblasts and collagen instead of having a significant cell loss. 

TGFβ has a central role in severe fibrotic diseases, including IPF, fibrotic heart diseases, systemic sclerosis, and diabetic nephropathy [52,53]. Stimulation with TGFβ in cardiac fibroblasts induces an increase in SCX expression [18], which was also observed by our group in lung fibroblasts (Appendix A). These data suggest that dysregulation of SCX expression in pathological fibrogenesis could be due to altered TGFβ production. Interestingly, TGFβ stimulation induces expression of other profibrotic genes, including α-SMA/ACTA2. Similarly, SCX/TCF3 overexpression can drive the elevated expression of α-SMA/ACTA2 independently of TGFβ. This information makes SCX a potential therapeutic target whose alteration would likely be less pleiotropic than targeting TGFβ.

It remains to be investigated why there are high levels of a transcription factor circulating in the blood, as they are usually compartmentalized in the cell nucleus to regulate gene expression. We know that SCX is stable enough in serum and produced in enough quantities to be detected by conventional ELISA tests. Because circulating TCF3 levels were high in healthy subjects and all patients, it is not a useful candidate biomarker. However, circulating SCX may be heterodimerizing with TCF3, promoting both factors stabilization and functionality. Furthermore, high circulating TCF3 levels could relate to high intracellular expression, a relevant condition for SCX functionality.

Nevertheless, we do not know yet if a specific cell receptor can recognize SCX and if its free serum form induces distinct cellular responses. Circulating SCX could be a reflection of cell death. Lung epithelial cells injury, activation, and apoptotic death are characteristic of pulmonary fibrosis [54]. Thus, it is also possible that SCX levels increase in serum due to cell damage and release of internal factors into the bloodstream.

SCX has been used as a marker for tendons and ligaments differentiation [37]. Here, we propose SCX as a candidate biomarker of severity of the fibrotic disease because high levels were found in severe cases of two fibrotic diseases. In contrast, low levels prevailed in healthy controls and HP, an inflammatory lung disease with few fibrotic cases. SCX as a biomarker will probably be more useful if applied together with other noninvasive IPF and SSc markers in a diagnostic-prognostic panel including chemokines (CCL18 and CXCL13), interleukins (Il-6, Il-13, and Il-8), and metalloproteinases (MMP7, MMP8, MMP9, MMP10, and MMP28) [7,55,56,57], once validated as fibrosis markers. 

SCX overexpression promoted elevated expression of target genes only when TCF3 was co-overexpressed. This result contrasts with previous reports where individual SCX overexpression induced increased expression of targets, including Twist, Snai1 [58], MMP2 [18], fibronectin 1 [19], and fibrillar collagens [21]. Because the cell context is critical in gain-of-function assays [59,60,61,62], we reasoned that cell cultures used in previous studies probably expressed adequate basal levels of TCF3, allowing the structuring of functional SCX/TCF3 heterodimers when SCX was overexpressed. Likewise, cell cultures used in the SCX publications above were derived from mice in comparison to the human cell cultures used in this work. Importantly, individual TCF3 overexpression did not promote elevated expression of αSMA/ACTA2, confirming the SCX requirement for fibrosing gene activation. 

SCX could be a therapeutic target, due to its confirmed involvement in the activation of fibrosing gene expression, together with the observed higher expression in cells, tissue, and serum derived from patients affected by severe fibrotic diseases. We suggest that the attenuation or blockade of its activity could be potentially useful as a novel therapeutic strategy. One of the advantages of targeting this molecule instead of TGFβ is that SCX and TCF3 work downstream from TGFβ, and their blockade would theoretically not alter the other pleiotropic and immunomodulatory activities of this molecule. Additional studies on this pathway should be performed.

## 4. Conclusions

This research is the first attempt to define high SCX expression in human lung and skin fibrosis. Our data suggest SCX as a candidate biomarker useful in identifying the severity of fibrotic diseases. This information could aid in the design of therapeutic strategies of suppression of SCX function, or in the construction of a noninvasive panel of relevant biomarkers of fibrotic diseases that will potentially enhance prognostic information for fibrotic pathologies. 

## 5. Materials and Methods

### 5.1. Study Population

Selected patients (Table 1) were male and female over 25 years old with a previous diagnosis of IPF (*n* = 56) or HP (*n* = 40) from the Instituto Nacional de Enfermedades Respiratorias (INER), Mexico. Patients did not show acute exacerbation of IPF, HP, or some other respiratory disease at the moment of sample collection. The diagnosis was made based on ATS/ERS/JRS/ALAT guidelines criteria [3]. 

Patients from the SSc cohort were recruited at the Department of Immunology and Rheumatology of the Instituto Nacional de Ciencias Médicas y Nutrición Salvador Zubirán (INCMNSZ). All patients were at least 18 years old at enrolment, and (*n* = 100) fulfilled the 2013 ACR SSc criteria [63]. At enrollment and every year, they underwent a standardized clinical evaluation that included the modified Rodnan Skin Score (mRSS) and all items included in the Medsger severity scale [64]. Results were recorded from the complete blood count, ESR, creatinine, liver function tests, urinalysis, spirometry, chest X-ray, or high-resolution chest tomography (HRCT), and transthoracic echocardiogram (TTE).

Patients were classified as diffuse (dcSSc) or limited (lcSSc) SSc according to the extent of skin involvement. If patients had skin involvement above the elbows or knees, which could include thorax or abdomen at any time in the course of the disease, they were classified as dcSSc. If the skin involvement was only distal to elbows and knees and never affected thorax or abdomen, the disease was classified as lcSSc. Early and late classification differed between dcSSc and lcSSc. Patients with lcSSc were considered in an early phase if they had up to five years of evolution of the disease from the first symptom attributable to the disease, while dcSSc patients were considered in an early phase if they had up to three years of evolution of the disease. 

Control groups were healthy males and females over 40 years old (*n* = 57) with no signs of respiratory disease that volunteered at the INER. The Ethics Committees at both, INER and INCMNSZ, approved the project and research protocol. All patients and control subjects were informed about the purpose of the study and signed informed consent to participate.

### 5.2. Blood Samples

Donated blood samples were collected in BD Serum Separator tubes (SST) at the time of diagnosis, with no previous treatment, for IPF, SSc, and HP. The IPF, HP, and healthy control samples belong to the biobank Interstitial Diseases Service at INER. SSc samples were obtained at different times during the natural history of the disease, and progression was classified as indicated above. The serum was separated from the clot by centrifuging at 2000× g for 10 min and frozen in aliquots until used.

### 5.3. SCX and TCF3 Quantification in Blood Serum

SCX and TCF3 concentrations in serum were determined by ELISA assays specific for the human scleraxis homolog A (SCXA) (Cloud-Clone Corp SEN 330Hu, Cloud-Clone Corp, Katy, TX, USA) and the human TCF3 (Abbexa abx250990, Abbexa, Cambridge, UK) following the instructions of the manufacturers. Duplicate undiluted samples were quantified, and SCX and TCF3 concentrations were calculated using standard curves for each assay. 

### 5.4. Cell Culture

Human primary fibroblasts from healthy and IPF were isolated from lung biopsies, as previously described [65]. Cells were maintained in growth medium (Ham’ s-F12, GIBCO Life Technologies, Carlsbad, CA, USA) supplemented with 10 % fetal bovine serum, penicillin (100 U/mL), and streptomycin (100 µg/mL) until 80% confluence was reached. Primary control cultures were named NOVA, whereas HIPF231, HIPF375, and HIPF397 were primary IPF cultures. Appendix A includes the donor characteristics of primary lung cell cultures. Human normal lung fibroblasts—CCD25-Lu and CCD8-Lu—from the American Type Culture Collection, were cultured in DMEM medium (GIBCO Life Technologies, Carlsbad, CA, USA) supplemented as above. All cell cultures were maintained in T-25 flasks at 37 °C in a humidified atmosphere of 5% carbon dioxide. CCD8-Lu fibroblasts for transduction experiments were grown as above, but the medium was supplemented with 10% newborn calf serum.

### 5.5. Real-Time PCR 

Total RNA from IPF tissues and cultured fibroblasts were extracted using the TRIzol^TM^ reagent (Invitrogen Life Technologies, Carlsbad, CA, USA). Two commercial human normal lung RNAs were used (Ambion AM7968, Ambion, Austin, TX, USA, and Origene HT1009, Origene, Rockville, MD, USA) as tissue controls to compare with RNA extracted from biopsies. RNA samples were reverse transcribed into cDNA using the RevertAid First Strand cDNA Synthesis Kit (Thermo Fisher Scientific, Waltham, MA, USA) following the manufacturer’s instructions. Real-time PCR was performed using LightCycler^®^ 480 System (Roche, Basel, Switzerland) with TaqMan probes (Thermo Scientific, Waltham, MA, USA) labeled with FAM (Hs03054634_g1 for SCX, Hs00164004_m1 for COL1A1, Hs01028970_m1 for COL1A2, Hs00426835_g1 for α-SMA/ACTA2, and Hs01012685_m1 for TCF3) and VIC (Hs02786624_g1 for GAPDH and Hs0280069S_m1 for HPRT). 

### 5.6. Western Blotting

Cultured fibroblasts were lysed with the RIPA buffer (Sigma Aldrich, St. Louis, MO, USA) containing a protease inhibitors cocktail (Calbiochem, Set V) and PMSF. After a 60 min incubation on ice, cells were sonicated, and cell debris was removed by centrifugation at 20,000× *g* for 20 min at 4 °C. Supernatants were collected, and protein concentrations were determined using the Bradford method. Forty micrograms of total extract was separated in 10–12% SDS-PAGE, transferred to polyvinylidene fluoride (PVDF) membranes (Bio-Rad, Hercules, CA, USA), and blocked with 5% milk in Tris-buffered saline with Tween 20 0.05% (TBS-T) for 1 h. Three micrograms of protein was used for SCX gain-of-function blots. Membranes were incubated overnight at 4 °C with primary antibodies: anti-SCX (1:600; Thermo Scientific PA5-23943, Waltham, MA, USA), anti-β-actin (1:200; Santa Cruz Biotechnology Inc. SC-47778, Dallas, TX, USA, or 1:10,000; Sigma A5441, St. Louis, MO, USA), anti-TCF3 (1:500; Thermo Scientific PA5-20900, Waltham, MA, USA), anti-Smooth Muscle Actin (1:500, Sigma A2547, St. Louis, MO, USA), and anti-vinculin (1:500, Santa Cruz Biotechnology Inc. SC-73614, Dallas, TX, USA). SCX in lung and skin tissue was detected with anti-SCX (1:200; Abcam ab58655, Cambridge, UK) after blocking with 5% BSA in TBS-T. Incubation with secondary antibodies coupled to horseradish peroxidase (anti-mouse and anti-rabbit IgG; Invitrogen Life Technologies, Carlsbad, CA, USA) was performed for 1 h at room temperature. The signal was visualized with the Super Signal West Pico Enhanced Chemiluminescence detection system (Thermo Scientific, Waltham, MA, USA) and normalized against β-actin or vinculin. Densitometric analyses were made with the Image Lab software 6.0 from Bio-Rad, Hercules, CA, USA. 

### 5.7. Immunohistochemical Staining

We examined the localization of SCX in three IPF and three healthy lung tissues. Immunohistochemical analyses were performed as described [66]. Briefly, paraffin-embedded lung tissues were obtained from biopsy or autopsy specimens of individuals with IPF or HP, and controls, in compliance with institutional review board-approved protocols. Three millimeter lung sections were dewaxed and rehydrated, and peroxidase activity was blocked with 3% H_2_O_2_ in methanol for 30 min, followed by a citrate buffer treatment to retrieve the antigen. The lung sections were incubated overnight at 4 °C with an anti-human SCX rabbit polyclonal antibody (1:25; Antibodies online ABIN9670006). A secondary biotinylated anti-immunoglobulin, followed by horseradish peroxidase-conjugated streptavidin (DAKO LSAB^®^2 System-HRP, Dako North America, Inc. K0609, Carpinteria, CA, USA) and the 3-amino-9-ethyl-carbazole (BioGenex, HK129-5K, San Ramon, CA, USA) substrate were used following the manufacturer’s instructions. Pictures of the hematoxylin-counterstained slides were acquired with a Nikon microscope equipped with the NIS-Elements AR software.

### 5.8. Adenoviral Vectors Construction and Transduction 

SCX and TCF3 infective adenoviral particles were prepared using the Adeno-X adenoviral system three (Clontech, Mountain View, CA, USA). Human SCX (0.6Kb) was subcloned from the cDNA clone SC316344 (OriGene, Rockville, MD, USA) into pAdenoX-DsRedExpress using the inFusion HD cloning kit (Clontech, Mountain View, CA, USA) and the following primers; adeno-SCXh-F: GTAACTATAACGGTCATGTCCTTCGCCACGCTGCG and adeno-SCXh-R: ATTACCTCTTTCTCCCCTCCTAACTGCGAATCGCTGTCT. Human TCF3 (E47) was amplified from cDNA derived from the CCD25 fibroblast line and cloned into the pCMV6-XL5 vector using the KpnI and XbaI restriction sites included in the following primers; KpnI-TCF3-F CCTCCTGGTACCAATGAACCAGCCGCAGAGG, and TCF3-XbaI-R CCTCCTTCTAGACGA GAGACACGGGACTTTTATAC. TCF3 (E47, 1.96Kb) was further subcloned into pAdenoX-DsRedExpress with the inFusion (Clontech) technology and the following primer pair; ad-TCF3-F GTAACTATAACGGTCATGAACCAGCCGCAGAGGATGGC and ad-TCF3-R: ATTACCTCTTT CTCCTCACATGTGCCCGGCGGGGTT. For packaging, PacI linearized recombinant adenoviral vectors (SCX, TCF3, and empty vector) were transfected (Lipofectamine) into Adeno-X 293 cells (Clontech, Mountain View, CA, USA). We used the Adeno-X-Maxi Purification Kit (Clontech, Mountain View, CA, USA) to purify adenoviral particles from a thirty P100 culture-dishes pellet. These cell cultures had reached 50% of the cytopathic effect. Adenoviral titers were calculated using the Clontech Adeno-X Rapid Titer Kit. Titers were as follows. Empty: 1.3 × 10^9^ infectious units per mL (IFU/mL); SCX: 1.82 × 10^9^ IFU/mL; and TCF3: 6.95 × 10^8^ IFU/mL.

CCD8Lu cells were seeded at 80% confluence on 6-well plates. After 24 h, cells were washed with PBS and transduced in DMEM media with no serum, using a Multiplicity of Infection (MOI) of 50 for each well. Cells were maintained under standard culture conditions for 48 h. The infection of the majority of the cells per well was verified using epifluorescence microscopy detecting the Red Fluorescent Protein (RFP) expression driven from all vectors. Then cells were washed with PBS, lysed, and harvested for protein or RNA, which were subsequently used for Western blot and quantitative PCR analyses, respectively.

### 5.9. Statistical Analyses

Results are presented as the median ± 95% CI for the SCX ELISAs and as the mean ± standard deviation (SD) for the TCF3 ELISAs. Statistical differences were analyzed by the one-way ANOVA test and the Tukey multiple comparison tests. Differences with probability values <0.05 were considered statistically significant. Data were analyzed using the statistical program GraphPad Prism 5.00 for Windows (GraphPad Software).

### 5.10. Ethics Approval and Consent to Participate

All subjects gave their informed consent for inclusion before they participated in the study. The study was conducted in accordance with the Declaration of Helsinki, and the protocol was approved by the Ethics Committees at INER (B15-14) and INCMNSZ (IRE-2557-18-20-1) approved this project and research protocols. All patients and control subjects were informed about the purpose of the study and signed informed consent to participate.

## Figures and Tables

**Figure 1 ijms-21-05012-f001:**
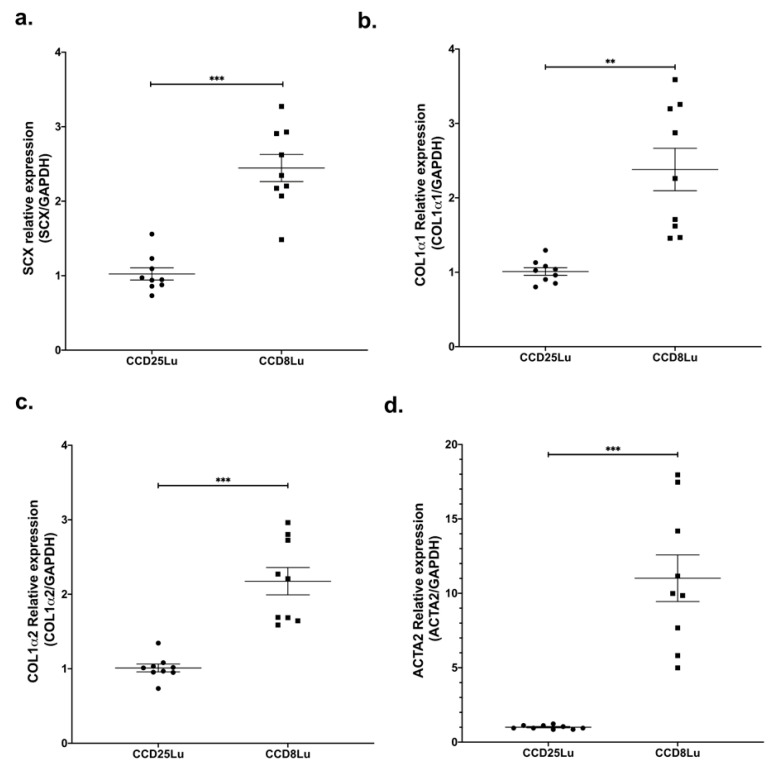
SCX expression levels in lung fibroblasts correlated with the expression of profibrotic genes. Expression of genes SCX (**a**), COL1A1 (**b**), COL1A2 (**c**), and α-SMA/ACTA2 (**d**) in pulmonary fibroblasts CCD25Lu and CCD8Lu. (a) Scleraxis expression was measured by qPCR using Taqman Hs03054634_g1 relative to GAPDH (Hs02786624_g1). Panels (b–d) correspond to COL1A1 (Hs00164004_m1), COL1A2 (Hs01028956_m1), and α-SMA/ACTA2 (Hs00426835_g1) expression relative to GAPDH, respectively. Statistical differences were assessed using the Student’s t-test. ** = *p* < 0.01, *** = *p* < 0.001.

**Figure 2 ijms-21-05012-f002:**
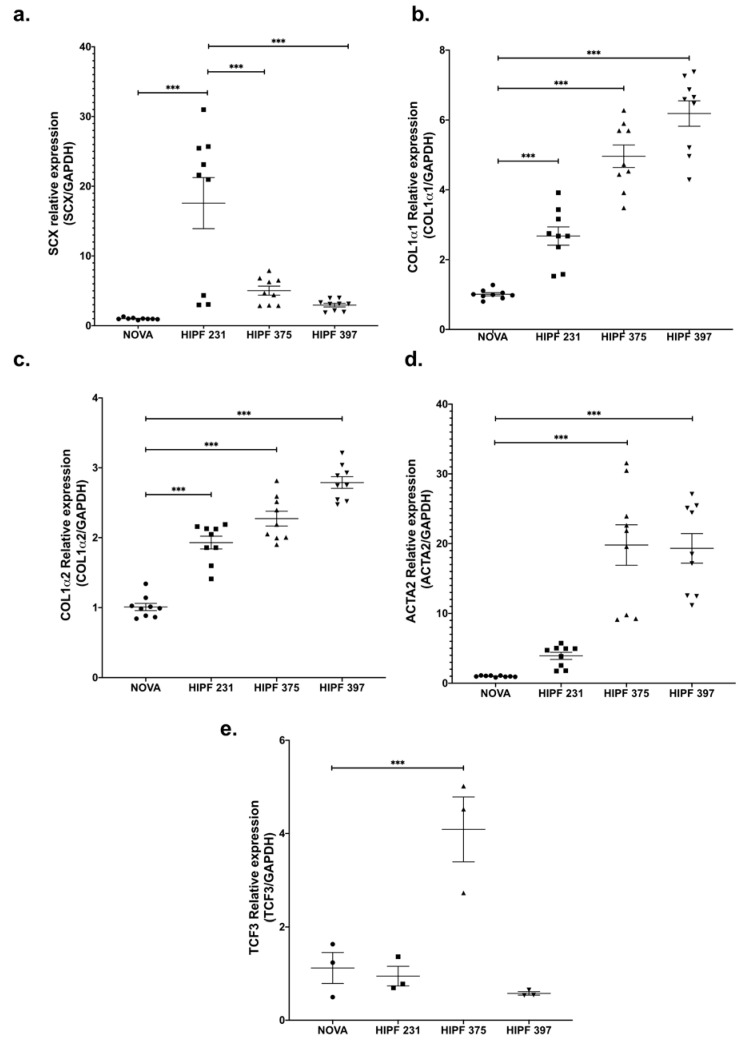
Gene expression of SCX and its target genes was high in IPF fibroblast primary cultures. mRNA levels of the indicated genes were quantified as in Figure 1 using cDNA samples from primary lung fibroblast cultures from a healthy donor (NOVA) and three IPF patients (HIPF). Relative expression of SCX (**a**), COL1A1 (**b**), COL1A2 (**c**), α-SMA/ACTA2 (**d**), and TCF3 (**e**). The statistical test used for this assay was one-way ANOVA. *** = *p* < 0.001.

**Figure 3 ijms-21-05012-f003:**
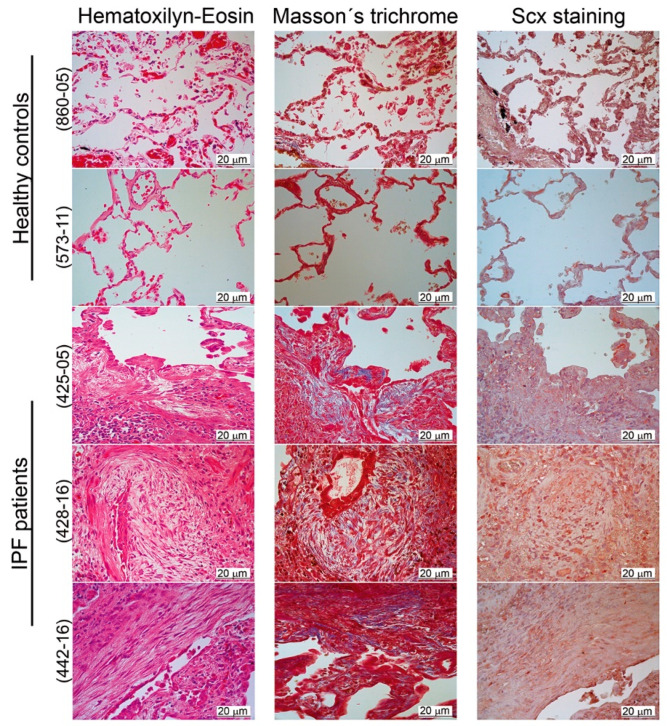
SCX expression was elevated in IPF tissue. Pulmonary tissue derived from IPF patients showed higher expression of SCX in comparison to healthy controls. Immunohistochemical analyses were made in two tissue samples from healthy controls (samples 573-11 and 860-05) and three IPF patients (425-05, 428-16, and 442-16). Samples were stained with hematoxylin–eosin and Masson’s trichrome. SCX was localized with a rabbit polyclonal antibody against human SCX (1:25; Antibodies online ABIN9670006). The fibroblastic foci in all IPF samples were localized in the center of each picture. In the hematoxylin–eosin stain, the fibroblastic foci were observed as pale areas composed of elongated cells, whose nuclei appeared stained with SCX (in the SCX stain). In the Masson’s trichome stain, the fibroblastic foci also showed blue coloring due to the presence of collagen.

**Figure 4 ijms-21-05012-f004:**
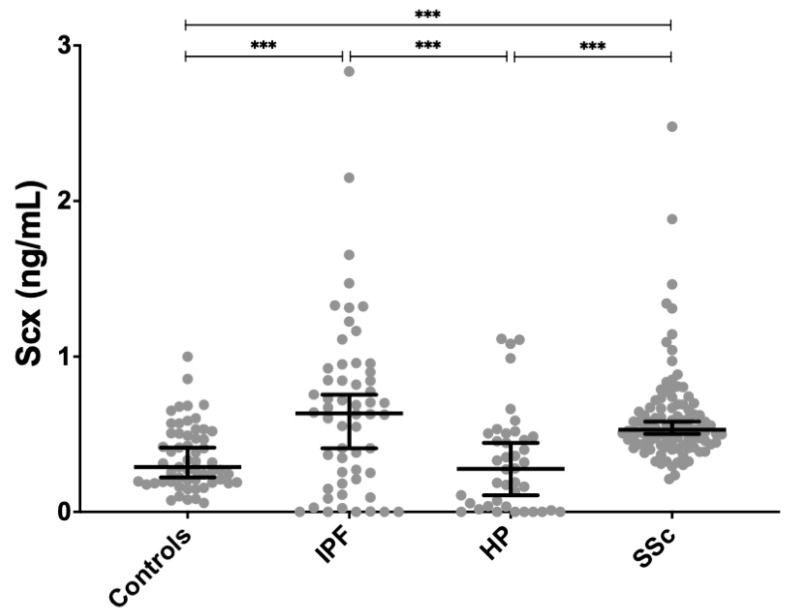
SCX serum concentrations were significantly higher in patients with IPF and SSc compared to controls. SCX serum levels in patients with HP were not different from controls but differed significantly from patients with IPF and SSc. *** = *p* < 0.001.

**Figure 5 ijms-21-05012-f005:**
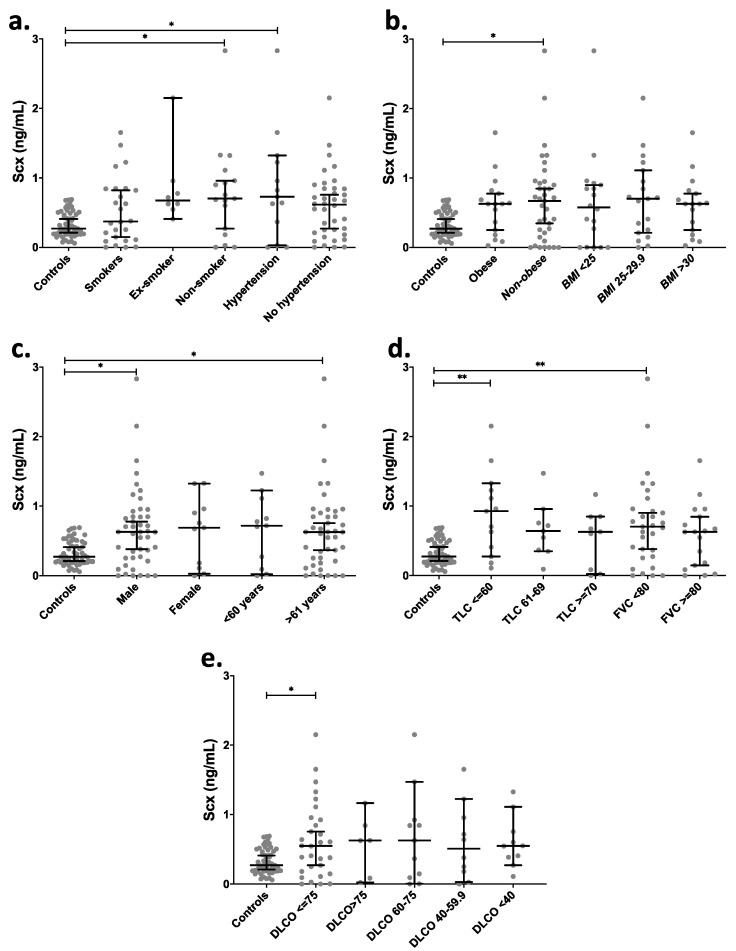
IPF patients with severely affected pulmonary function showed higher levels of circulating SCX. SCX serum levels grouped by (**a**) smoking habit and hypertension, (**b**) BMI, (**c**) sex and age, (**d**) TLC and FVC, and (**e**) DLCO. Appendix A includes an excel file with raw data. Higher SCX serum levels were found in patients with severely diminished TLC (≤ 60%) and with FVC below 80%. Ex-Smoker: patient that has not smoked for at least one year. Non-Smoker: patient with no smoke habit. Hypertension: values above 139 mmHg in systolic pressure and 89 mmHg in diastolic pressure were considered as hypertension [28]. * = *p* < 0.05, ** = *p* < 0.01. Total Lung Capacity (TLC): maximum volume of air present in the lungs. A normal person can inhale approximately 6 L. Forced Vital Capacity (FVC): volume of air that can be forcibly blown out after a full inspiration. Normal values are approximately 4.6 L [29].

**Figure 6 ijms-21-05012-f006:**
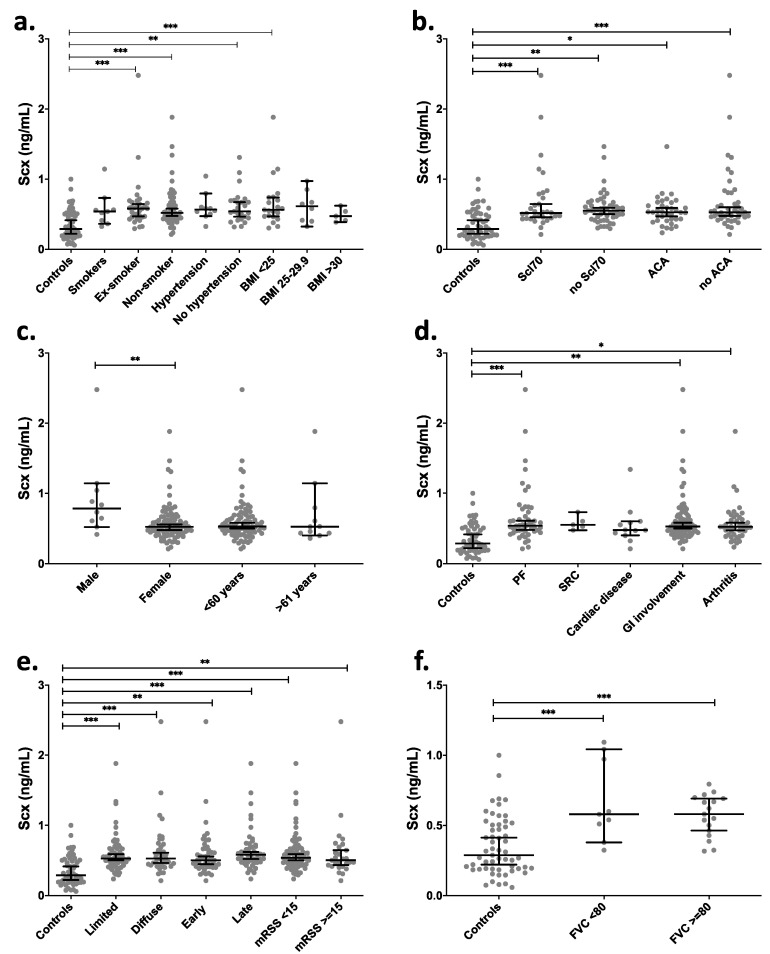
SCX serum levels were high in late progression SSc patients and patients suffering associated pulmonary fibrosis. SCX circulating levels were grouped by (**a**) smoking habit and BMI, (**b**) serologic profile, (**c**) sex and age, (**d**) internal organ involvement, (**e**) classification and progression, and (**f**) FVC. Appendix A includes an excel file with raw data. Secondary organ involvement comprised pulmonary fibrosis (PF), scleroderma renal crisis (SCR), and gastrointestinal involvement (GI). The GI included esophageal dysmotility, esophageal stricture, small bowel hypomotility or dilation, small bowel bacterial overgrowth, malabsorption syndrome, or use of parenteral nutrition. (d) SSc classification and progression. * = *p* < 0.05. ** = *p* < 0.01. *** = *p* < 0.001. Scl70 = anti-topoisomerase I antibodies.

**Figure 7 ijms-21-05012-f007:**
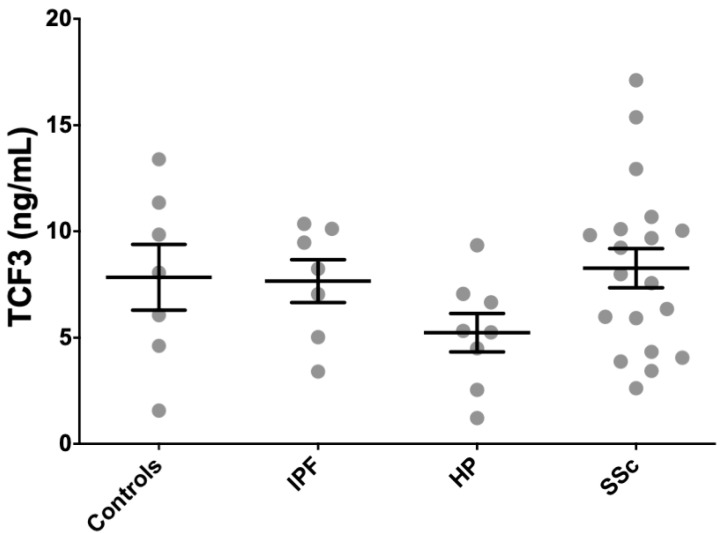
TCF3 serum levels did not associate with fibrotic illness. TCF3 serum levels in all patients and healthy subjects were similarly high.

**Figure 8 ijms-21-05012-f008:**
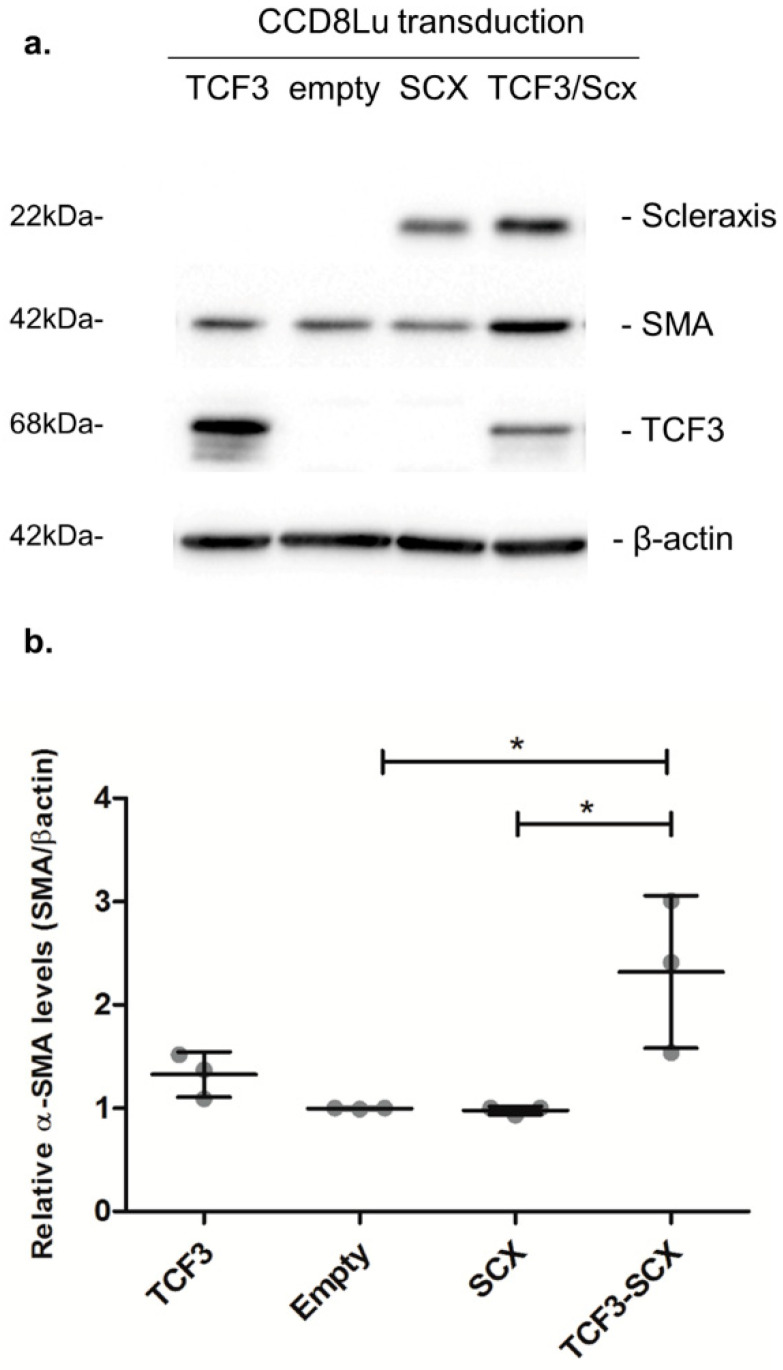
SCX overexpression promoted an induction in α-SMA expression that was dependent on TCF3. (**a**) The expression of the indicated proteins was analyzed using immunoblots. CCD8Lu cells were transduced with adenoviral vectors for 48 h. The infective particles included an empty vector as a control, and adenoviral vectors encoding SCX, TCF3, and a combination of SCX and TCF3. (**b**) Densitometric analyses of α-SMA induction relative to β-actin, quantifying three independent samples of each transduction. A one-way ANOVA test confirmed statistical differences in α-SMA expression in empty vs. TCF3-SCX samples. * *p* = 0.0086.

**Table 1 ijms-21-05012-t001:** Patients and controls characteristics.

	Controls	IPF	HP	SSc
Gender (M/F)	15/42	45/11	9/31	10/90
Average age (Years)	62 ± 7	66 ± 11	57 ± 12	45 ± 12
Age range	41–78	50–78	30–78	26–72
Smoking status (never/past/current)	28/0/14	17/8/26	30/5/5	58/32/9
Hypertension (positive/negative)	4/38	13/38	16/22	10/25
TLC (% of predicted value)	105 ± 13	64 ± 16	62 ± 22	-
FVC (% of predicted value)	91 ± 19	72 ± 21	67 ± 25	86 ± 34
DLCO (% of predicted value)	-	54 ± 23	46 ± 25	-
Lymphocytes BAL (%)	-	10 ± 9	40 ± 25	-
Macrophages BAL (%)	-	88 ± 33	58 ± 29	-
SSc classification (limited/diffuse) (limited/diffuse)	-	-	-	57/43
SSc progression (early/late)	-	-	-	51/49
mRSS	-	-	-	10 ± 9

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
