# Peer review of "The Transcription Factor SCX is a Potential Serum Biomarker of Fibrotic Diseases"

_ijms, 2020, doi:10.3390/ijms21145012_

Round 1

Reviewer 1 Report

This manuscript builds on data from heart fibrosis which show that defects in SCX function can lead to cardiac fibrosis and SCX downregulation reduces the number of cardia myofibroblasts and attenuates myofibroblasts.

Major suggestions:

The authors should explain why SCX still has anti-fibrotic effects if it requires increased TCF3 as shown in Figure 8, yet IPF patients and SSc patients do not have elevated TCF3 levels.

How do the authors explain the fact that a transcription factor, which should be localized to the nucleus of cells, is found in the circulation? Is that a reflection of cell death?

Figure 1 compares levels in only two control fibroblasts strains. Several should be included due to variability in human cells, especially since one donor is 7 years old and the other donor is 48 years old.

Data shown in Figure 2 compare expression levels in three IPF fibroblast strains compared to one non-IPF control. Since there is variability in human cells, n=1 normal control is insufficient and additional samples should be included.

It is surprising that the levels of SCX are inversely correlated with levels of Col1A1, Col1A2, and ACTA2. One would expect that as levels of a pro-fibrotic factor increase, so do the levels of its target genes.

Figure 3 data are derived from 2 controls. Using =2 is inadequate with statistical analyses.

If SCX is a biomarker for fibrosis, then analysis of SSc patients should be based on presence/absence of ILD and limited skin involvement vs diffuse skin involvement to truly show that those with more fibrosis (higher skin score) have higher levels of SCX. However, the data show SCX levels are the same irrespective of these clinical variables which is surprising if SCX is truly a biomarker of fibrosis.

Table 1 is missing lung function data from SSc patients.

How were the subsets of patients selected for Figure 7 for the measurement of TCF3?

When comparing multiple conditions, such as Figure 8, the student’s t-test is not the appropriate statistical method. ANOVA with appropriate correction should be used. Some figure legends state the t test was used while methods state ANOVA was used.

Data in Figure 8 are again generated with a single fibroblast line.

It is incorrect to state that patients with Scl70/topoisomerase I antibodies all have lung disease.

Figure S4, the authors should not state that SCX is increased in bleomycin lung since only two controls were examined and they had very variable levels.

N=2 is insufficient for figure S6 as well.

Minor suggestions:

Some of the references don’t seem to mention the statements they are supposed to refer to.

The manuscript requires editing by a native English speaker.

Reviewer 2 Report

Comments to the manuscript entitled "The transcription factor SCX is a potential serum biomarker of fibrotic diseases“ by Ramirez-Aragon et al

Ramirez-Aragon and coworkers reseached the potential function of SCX as a potential serum biomarker for fibrotic (lung) diseases. I have a few comments.

  1. The authors should research if SCX is induced by TGF-beta1. For this, human lung fibroblast cell lines should be analysed for SCX gene- and protein expression in response to TGF-beta1 exposure/incubation. In addition, the authors can also research the expression pattern of TCF3. This experiment would add to the story.
  2. The authors performed statistics. However, the figures 1 and 2 including the subfigures lack the n-number of experiments/replicates. How were the means calculated?

  3. Figure 8a: The authors should provide the original uncropped and unadjusted blots from all immunoblot results shown in Figure 8a, including the molecular size markers/molecular size standard [in kDa], in the supplementary files.

  4. Figures 1, 2, and 8b: All bar graphs should be replaced by dot graphs to show distribution of values and number of biological replicates.

  5. The cloning procedures for SCF and TCF cDNAs into adenoviral vectors are not adequately described. (amplification primers, restriction sites, size of cloned cDNA fragments of SCF and TCF)

  6. Unfortunately, the supplementary files could not be opened and seen by this reviewer. Something is not okay with these files!

Reviewer 3 Report

This article indicated SCX could be a biomarker and potential therapeutic target for IPF and other fibrotic lung diseases.

Major Comments

1.Supplementary file 2.

IPF: The patient of ILD 2332 had BAL lymphocyte of 87% and the age is only 38 years. Is he truly diagnosed as IPF by multidisciplinary discussion (MDD) according to international guideline? The reviewer also has slight doubts in the patient of Z-468, and ILD 2319 whose BAL lymphocytosis (> 30%).

2. Page 11, Line 244-246, Figure 7.

In this study, serum SCX levels were measured in 57 healthy controls, 57 IPF, 39 HP and 100 SSc patients. Are there any reasons for the small number of TCF3 measurements?

3. Page 13, Line 292-302

SCX heterodimerizes with TFC3, what is the mechanism by which SCX and TCF3 form or do not form heterodimerization?

Minor Comments

1. Page 3, Line 105-106: Does this sentence refer to Figure 2a? The notation in Figure 2a is missing in this sentence.

2. Page 8, Figure 5:

Figure 5A: SCX levels showed significant increase in ex-smokers and hypertension patients in contrast with the control group in the text. Figure 5A shows that it is significantly different from non-smoker.

Line 195:  *=p<0.05 is missing in the sentence.

3. Page 17, Line 516. Results are presented as the mean ± standard deviation (SD). If it is not normally distributed, it is better to show it by the median value rather than the mean value.

Round 2

Reviewer 1 Report

Incomplete revision

Reviewer 2 Report

Ramirez-Aragon and co-authors have revised their paper appropriately. I have nothing to add.

Reviewer 3 Report

In this revised version of the manuscript, the authors have carefully addressed all of the points I raised.